# Reverse Stable Diffusion: What prompt was used to generate this image?

## Abstract

Text-to-image diffusion models such as Stable Diffusion have recently attracted the interest of many researchers, and inverting the diffusion process can play an important role in better understanding the generative process and how to engineer prompts in order to obtain the desired images. To this end, we introduce the new task of predicting the text prompt given an image generated by a generative diffusion model. We combine a series of white-box and black-box models (with and without access to the weights of the diffusion network) to deal with the proposed task. We propose a novel learning framework comprising of a joint prompt regression and multi-label vocabulary classification objective that generates improved prompts. To further improve our method, we employ a curriculum learning procedure that promotes the learning of image-prompt pairs with lower labeling noise (*i.e.* that are better aligned), and an unsupervised domain-adaptive kernel learning method that uses the similarities between samples in the source and target domains as extra features. We conduct experiments on the DiffusionDB data set, predicting text prompts from images generated by Stable Diffusion. Our novel learning framework produces excellent results on the aforementioned task, yielding the highest gains when applied on the white-box model. In addition, we make an interesting discovery: training a diffusion model on the prompt generation task can make the model generate images that are much better aligned with the input prompts, when the model is directly reused for text-to-image generation. Our code is publicly available for download at https://github.com/anonymous.

## 1 Introduction

In computer vision, diffusion models (Croitoru et al., 2023; Song et al., 2021b; Ho et al., 2020; Song & Ermon, 2020; 2019; Sohl-Dickstein et al., 2015; Dhariwal & Nichol, 2021; Nichol & Dhariwal, 2021; Song et al., 2021b) have recently emerged as a powerful type of deep generative models. Their widespread adoption can be attributed to the impressive quality and variety of the generated samples. These models have been utilized for a range of tasks, including unconditional image generation (Song et al., 2021b; Ho et al., 2020; Song & Ermon, 2020), super-resolution (Saharia et al., 2023; Daniels et al., 2021; Rombach et al., 2022; Chung et al., 2022), image editing (Avrahami et al., 2022), text-to-image generation (Rombach et al., 2022; Ramesh et al., 2022; Saharia et al., 2022), and many others (Croitoru et al., 2023). Among these applications, text-to-image generation has garnered significant attention due to the exceptional ability of diffusion models to generate diverse high-quality images that seamlessly align with the input text. The Stable Diffusion model (Rombach et al., 2022) has stood out as a highly effective approach for achieving impressive results in text-to-image generation. This has been accomplished by incorporating diffusion processes within a low-size semantic embedding space, producing faster inference time, while maintaining similar or even better outcomes than other state-of-the-art techniques. The image synthesis is conditioned on the text by encoding and integrating the text in the cross-attention mechanism. Although the use of diffusion models has yielded positive results in text-to-image generation, there is a notable lack of research regarding the understanding of these models. For example, there is a rising need to

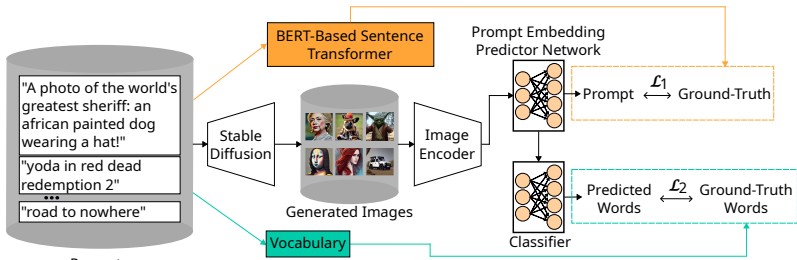

Figure 1: Our learning framework for prompt embedding estimation, along with its vocabulary classification task. We transform the input prompts via a sentence transformer for the embedding estimation task and we use a vocabulary of the most common words to create the target vectors for the classification task.

understand how to design effective prompts that produce the desired outcome. Further investigation in this area will be crucial for the advancement of text-to-image generation models. For instance, although the generated images are usually correlated with the input prompts, there are many situations when certain tokens from the text prompt are completely ignored, or some latent properties of the respective tokens are not included.

To this end, our work focuses on introducing a novel task that involves reversing the text-to-image diffusion process. Due to its notoriety, we particularly focus on reversing the Stable Diffusion model. Given an image generated by Stable Diffusion, the proposed task is to predict a sentence *embedding* of the original prompt used to generate the input image. This formulation allows us to avoid measuring text generation performance with problematic text evaluation measures (Callison-Burch et al., 2006). To address the image-to-text generation task under the new formulation, we utilize image encoders to extract image representations, which are subsequently projected to the sentence embedding space via fully connected layers. As underlying models, we consider three state-of-the-art architectures that are agnostic to the generative mechanism of Stable Diffusion, namely ViT (Dosovitskiy et al., 2021), CLIP (Radford et al., 2021) and Swin Transformer (Liu et al., 2021). We also include the U-Net model from Stable Diffusion, which operates in the latent space. Notably, we consider both black-box and white-box models (with and without access to the weights of the diffusion network), showing that our novel approach achieves good performance regardless of the underlying architecture. The first novel component in our training pipeline is an extra classification head, which learns an additional task, that of detecting the most common words in the vocabulary used to write the training prompts. This classification task constrains our model to produce more accurate text prompts. In Figure 1, we illustrate this component and our learning method to reverse the text-to-image generative process. Additionally, our second component is an innovative curriculum learning method based on estimating the difficulty of each training sample via the average cosine similarity between generated and ground-truth text prompts, measured over several training epochs. Measuring the cosine similarity over several epochs before employing the actual curriculum gives us a more objective measurement of sample difficulty scores, not influenced by the curriculum or the state of the neural model. Our third component aims to adapt the model to the target domain via a kernel representation. This component harnesses unlabeled examples from the target domain to compute second-order features representing the similarities between samples from the source (train) and target (test) domains. We call this component the domain-adaptive kernel learning (DAKL) framework, and we apply it as a meta-model on the median embedding of the ViT, CLIP, Swin-T and U-Net models.

We carry out experiments on the DiffusionDB data set (Wang et al., 2022), after filtering out examples with near duplicate prompts. We report comparative results to illustrate the influence of our training pipeline. Moreover, we empirically study the impact of our prompt generation task on the correlation between generated images and input prompts, during text-to-image generation with Stable Diffusion. Notably, our findings reveal that training the diffusion model on the image-to-text prediction task, then reintegrating the model to perform the original task, text-to-image generation, can significantly improve the alignment between the generated images and the input prompts.

In summary, we make the following contributions:

- We propose the new task of image-to-text-embedding prediction to study the possibility of reversing the text-to-image generation process of diffusion models.
- We propose a training pipeline comprising three novel components (a classification head, a curriculum learning method, and a domain-adaptive kernel learning framework) and demonstrate its usefulness on four underlying models.
- We showcase a promising application of prompt generation models in text-to-image generation, indicating that a diffusion model trained on generating text prompts can also be used to generate images that are better aligned with their prompts.

## 2 Related Work

As the proposed task is novel, there is no specific research on the subject. Yet, one possible approach would be to use an image captioning model to generate a natural language description of an image generated by Stable Diffusion. Next, we can employ a sentence transformer to obtain the prompt embedding. Although our task is slightly different, since the final output is a prompt embedding, we consider image captioning methods (Stefanini et al., 2023) as related work. The earliest deep learning approaches for image captioning (Vinyals et al., 2015; Karpathy & Li, 2015; Mao et al., 2015; Fang et al., 2015; Jia et al., 2015) used CNNs (Krizhevsky et al., 2012; Simonyan & Zisserman, 2014; Szegedy et al., 2015) as high-level feature extractors, and RNNs (Hochreiter & Schmidhuber, 1997) as language models to generate the description conditioned on the extracted visual representations. Further developments on this topic focused on improving both the visual encoder and the language model (Xu et al., 2015; Lu et al., 2017; Dai et al., 2018; Chen et al., 2018; Wang et al., 2017; Gu et al., 2018; Yang et al., 2019; Li et al., 2019; Herdade et al., 2019; Huang et al., 2019; Pan et al., 2020). A major breakthrough was made by Xu et al. (2015), who, for the first time, incorporated an attention mechanism in the captioning pipeline. The mechanism allowed the RNN to focus on certain image regions when generating a word. Thereafter, other attention-based models were proposed by the research community (Lu et al., 2017; Huang et al., 2019; Pan et al., 2020).

Recent image captioning models have adopted the transformer architecture introduced by Vaswani et al. (2017). Most studies (Herdade et al., 2019; Guo et al., 2020; Luo et al., 2021) used the original encoder-decoder architecture, followed closely by architectures (Li et al., 2020; Zhang et al., 2021) inspired by the BERT model (Devlin et al., 2019). Another research direction leading to improvements on the image captioning task is focused on multimodal image and text alignment. This was usually adopted as a pre-training task (Radford et al., 2021; Li et al., 2022). The CLIP (Contrastive Language-Image Pre-training) model introduced by Radford et al. (2021) represents a stepping stone in the multimodal field, showing great zero-shot capabilities. It involves jointly training two encoders, one for images and one for text, with the objective of minimizing a contrastive loss, *i.e.* whether or not the image and the text match. As a result, the cosine similarity between the embedding of an image and that of its description, resulting from the corresponding encoders, should be high, while counterexamples produce a low similarity. Shen et al. (2022) and Mokady et al. (2021) studied the impact of the CLIP visual encoder on the image captioning task, without additional fine-tuning. The BLIP model (Li et al., 2022), short for Bootstrapping Language-Image Pre-training, introduces a novel architectural design and pre-training strategy. Its aim is to create a model that can both understand and generate language.

Our framework is distinct from image captioning approaches because we do not aim to generate natural language image descriptions, but rather to map the images directly into a text embedding space. Moreover, we propose three novel contributions and integrate them into a unified learning framework to significantly boost the performance of the employed models, when these models are used both independently and jointly.

## 3 Method

**Overview.** Our approach for the image-to-prompt-embedding generation task employs four image encoders to extract image representations. These representations are then converted into sentence embeddings using dense layers. To train these models, we use a novel training pipeline that integrates multiple novel components. An overview of the

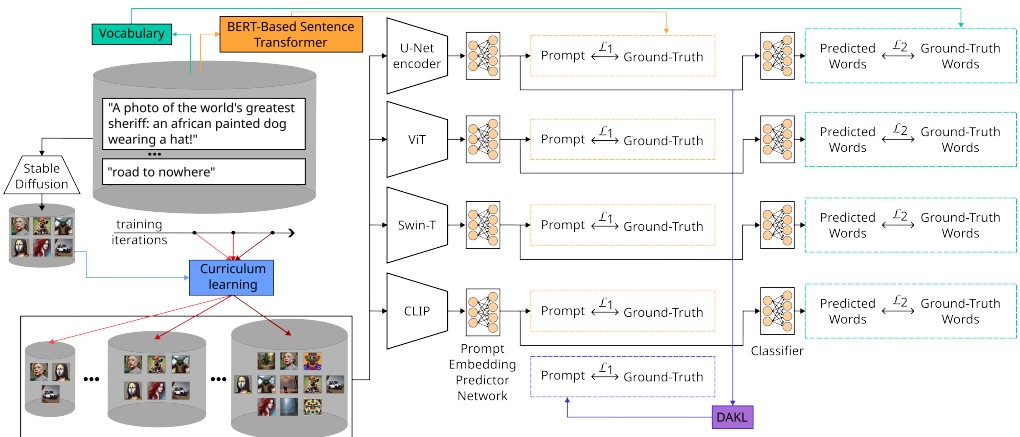

Figure 2: Our prompt embedding estimation framework, which includes a classification task, a curriculum learning procedure and a domain-adaptive kernel learning (DAKL) method. As part of the curriculum learning procedure, we start with a small set of easy examples and gradually add more complex samples in the subsequent iterations. We transform the input prompts via a sentence transformer for the embedding estimation task, and we use a vocabulary of the most common words to create the target vectors for the classification task. Lastly, we apply the DAKL method on the median embedding of the four models to further improve the performance of the framework.

proposed training pipeline is shown in Figure 2. We train our model using two objectives. The main objective estimates the cosine distance between the predicted and the target sentence embeddings. The second objective aims to accomplish a classification task by predicting whether the words from a predefined vocabulary are present in the original prompt, given the generated image. Our pipeline employs a curriculum learning strategy, filtering out complex examples in the initial training iterations. As the training process continues, we gradually increase the complexity threshold, until we encompass all training samples. Finally, our training pipeline embeds a domain-adaptive kernel learning method to boost the performance of the overall framework.

**Main objective.** Our goal is to understand the generative process of text-to-image diffusion models. In particular, we choose one of the most representative diffusion models to date, namely Stable Diffusion (Rombach et al., 2022). We argue that reversing the process, performing image-to-text generation, plays an important role in this regard. However, we do not focus on generating natural language descriptions of images. Instead, we concentrate on training models to predict embedding vectors as similar as possible to the actual embeddings obtained by applying a sentence transformer (Reimers & Gurevych, 2019) on the original prompts. We solve this indirect task to avoid the use of the problematic text evaluation measures (Callison-Burch et al., 2006) commonly employed in image captioning, such as BLEU (Papineni et al., 2002). Therefore, we map generated images to vectors that reside in the embedding space of a sentence transformer (Reimers & Gurevych, 2019), which correspond to the actual prompts. Chen et al. (2023) found that BERT-style encoders are better than CLIP-style encoders, when it comes to encoding text. We thus consider that the sentence BERT encoder generates more representative target embeddings for our task. Formally, let $(x_i, y_i)_{i=1}^n$ denote a set of $n$ training image and prompt pairs. Let $f_\theta$ and $s$ be a prompt generation model and a sentence transformer, respectively. Then, the main objective is to minimize the following loss:

$$\mathcal{L}_1 = \frac{1}{n} \sum_{i=1}^n \left( 1 - \frac{\langle f_\theta(x_i), s(y_i) \rangle}{\|f_\theta(x_i)\| \cdot \|s(y_i)\|} \right), \tag{1}$$

where $\theta$ represents the learnable parameters of the model.

**Multi-label vocabulary classification.** Although the objective defined in Eq. (1) maximizes the cosine similarity between the output and the target embedding, the training process does not directly exploit the prompt content in this setting. Therefore, we introduce an additional classification task, in which the model has to analyze the generated input image

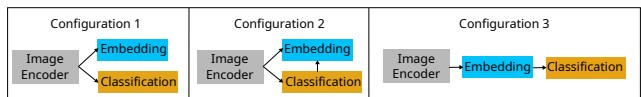

Figure 3: Configurations for the classification and embedding prediction heads. In the first configuration, the heads are separate, being fed with the same features. In the second configuration, the output of the classification head is concatenated with the image encoding to create the final intake for the embedding prediction head. In the third configuration, the classification is carried out using the predicted embedding as input.

and determine if each word in a predefined vocabulary is present or not in the original prompt used to generate the image. We select the most frequent adjectives, nouns and verbs as elements of the vocabulary. Formally, we create a vocabulary $V = \{t_1, \ldots, t_m\}$ and, for each training prompt $y_i$, we build a binary vector $l_i = (l_{i1}, \ldots, l_{im})$ as follows:

$$l_{ij} = \left\{ \begin{array}{l} 1, \text{if } t_j \in y_i \\ 0, \text{otherwise} \end{array} \right., \forall j \in \{1, ..., m\}. \tag{2}$$

To learn the frequent vocabulary words, we add an additional classification head, containing $m$ neurons, thus having one neuron per vocabulary item. Given a training set of $n$ image and prompt pairs $(x_i, y_i)_{i=1}^n$, we minimize the following objective with respect to $\hat{\theta}$:

$$\mathcal{L}_2 = \frac{1}{n \cdot m} \sum_{i=1}^{n} \sum_{j=1}^{m} l_{ij} \cdot \log\left(1 - \hat{l}_{ij}\right) + (1 - l_{ij}) \cdot \log\left(\hat{l}_{ij}\right), \tag{3}$$

where $\hat{l}_i = f_{\hat{\theta}}(x_i)$ is the vector of probabilities for all vocabulary items, and $\hat{\theta}$ consists of the parameters of the new classification head, as well as the trainable parameters $\theta$ of the underlying model. We combine the losses from Eq. (1) and Eq. (3) into a single term:

$$\mathcal{L} = \mathcal{L}_1 + \lambda \cdot \mathcal{L}_2, \tag{4}$$

where $\lambda$ is a hyperparameter that controls the importance of the classification task with respect to the prompt estimation task.

An important aspect for the classification task is the position of the new head relative to the head responsible for generating the prompt embedding, where each head is a single fully connected layer. We study three possible configurations, as illustrated in Figure 3. In the first configuration, the vocabulary prediction head is added at the same level as the prompt embedding head, as an independent head, which can only influence the preceding representations of the model. In this configuration, the classification head can indirectly influence the prompt generation head. In the second configuration, the output of the classification head is passed as input to the embedding prediction head. This configuration aims to improve the prompt embedding by allowing the model to see which words are likely to be present in the prompt. In the third configuration, the model is constrained to produce an embedding from which the active words can be recovered. This configuration can better align the generated prompt with the actual prompt, when words from the vocabulary are present in the original prompt. In the experiments, we empirically compare these three configurations.

**Curriculum learning.** If there is a considerable amount of examples with noisy labels in the training data, the weights of our model could converge to suboptimal values during training. Since the input images are generated by a neural model, *i.e.* Stable Diffusion, we naturally expect to have images that do not fully represent the text prompts. Therefore, to make the training of the prompt generation models robust to noisy labels, we propose to employ a curriculum learning technique (Bengio et al., 2009). Curriculum learning is a method to train neural models inspired by how humans learn (Soviany et al., 2022). It involves organizing the data samples from easy to hard, thus training models on gradually more difficult samples. In our case, we propose to train the neural networks on samples with progressively higher levels of labeling noise. In the beginning, when the weights are randomly initialized, feeding easier-to-predict (less noisy) examples, then gradually introducing harder ones, can help convergence and stabilize training.

Given that our outputs are embeddings in a vector space, we harness the cosine similarity between the generated prompt and the ground-truth prompt during training. Thus, we

propose a two-stage learning procedure. In the first phase, we train the network for a number of epochs using the conventional learning regime, storing the cosine similarity of each sample after every epoch. The only goal of the first (preliminary) training phase is to quantify the difficulty of learning each training sample. The difficulty score of a training example is computed as the mean of the cosine similarities for the respective example. Generalizing the observations of Swayamdipta et al. (2020), we conjecture that the resulting difficulty score is proportional to the amount of noise, essentially quantifying the level of misalignment between the input image $x_i$ and the corresponding text prompt $y_i$. To support our conjecture, we show examples of easy, medium and hard images in the supplementary (see Figure 3). For the second training phase, we reset the training environment and split the training data into three chunks, such that each chunk represents a difficulty level: easy, medium, and hard. Finally, we train the model again until convergence in the same number of steps as before, by gradually introducing each data chunk, starting with the easiest one and progressing to the hardest one. The model still gets to see the whole data set, but it ends up spending less time learning noisy examples.

We consider two alternative data splitting heuristics. The first one is to divide the data set into three equally-sized chunks, inspired by Ionescu et al. (2016). The second splitting heuristic involves setting two threshold values for the cosine similarity and splitting the data set according to these thresholds. As far as the latter heuristic is concerned, we propose a strategy for optimally choosing the threshold values. The first threshold should be set such that the first data chunk, *i.e.* the easy-to-predict samples, contains a significant portion of the data set. The second threshold should be sufficiently low to only include a small subset of the data set, *i.e.* the really difficult examples.

**Domain-adaptive kernel learning.** To further improve the performance of the employed models, we introduce a domain-adaptive kernel method that allows the models to exploit information about the target (test) domain without using the ground-truth embeddings. For a given similarity function $k$, we compute the kernel matrix $K$ containing similarities between pairs of samples from both source (training) and target (test) domains. Thus, we have that:

$$K_{ij} = k(z_i, z_j), \forall i, j \in \{1, \ldots, n + p\}, \tag{5}$$

where $Z = \{z_1, \ldots, z_{n+p}\}$ denotes the union of the training set $X$ with an unlabeled data set $\bar{X}$ from the target domain, *i.e.* $Z = X \cup \bar{X} = \{x_1, \ldots, x_n, \bar{x}_1 \ldots \bar{x}_p\}$. Note that $n$ and $p$ indicate the number of examples in $X$ and $\bar{X}$, respectively. For simplicity, we use the linear kernel as the function $k$. Next, we normalize the matrix $K$ and utilize the radial basis function (RBF) to create a fresh set of second-order features for every sample, $\{z_i\}_{i=1}^{n+p}$. These features reflect the similarities between each sample and all other samples in the data set, including those present in the set representing the target domain. Formally, the kernel normalization can be performed via:

$$\hat{K}_{ij} = \frac{K_{ij}}{\sqrt{K_{ii} \cdot K_{jj}}}, \tag{6}$$

while the RBF transformation is given by:

$$K_{\text{DA}} = \exp\left(-\gamma\left(1 - \hat{K}_{ij}\right)\right), \tag{7}$$

where $i, j \in \{1, \ldots, n + p\}$. To predict the prompt embeddings, we reapply the linear kernel on the revised features contained by $K_{\text{DA}}$ and train a dual regression model based on $L_2$ regularization. The final output of the regression model constitutes the improved prompt embedding of our framework. This approach offers a significant benefit, namely that the model has the opportunity to gather crucial information about the target domain in an unsupervised manner, by harnessing the similarities between source and target domain samples. However, kernel methods are inefficient on large-scale data sets, due to their quadratic dependence on the number of training samples, which can lead to unreasonably high memory requirements. This statement directly applies to our work, as trying to compute a kernel matrix on our training set from DiffusionDB (even after filtering out near duplicates) produces an out-of-memory error (on a machine with 256 GB of RAM). To solve this issue, we employ the k-means algorithm to extract a set of centroids, denoted as $C = \{c_1, \ldots c_r\}$, from our training set. Then, we substitute the original training set $X$ with this new set $C$ in the aforementioned method, which is significantly more efficient.

**Underlying models.** To demonstrate the generalization power of our training framework to multiple pre-trained neural networks, we employ the proposed framework on four distinct deep architectures as image encoders: ViT (Dosovitskiy et al., 2021) (Vision Transformer), CLIP (Radford et al., 2021) (Contrastive Language-Image Pre-training), Swin Transformer (Liu et al., 2021), and the encoder of the U-Net used in Stable Diffusion (Rombach et al., 2022). The former three networks represent black-box models with no access to the weights of Stable Diffusion, while U-Net comes with the weights from Stable Diffusion, being a white-box approach. ViT (Dosovitskiy et al., 2021) is one of the first computer vision models based on the transformer architecture. It divides the images into fixed-size patches and treats them as input tokens, enabling the model to learn relationships between patches. We employ the base variant, namely ViT-Base. CLIP (Radford et al., 2021) is a groundbreaking model that learns to understand images and text jointly, using a contrastive framework. By leveraging a large data set of paired images and their corresponding textual descriptions, CLIP learns to associate images and captions in a shared embedding space. In our case, we take only the visual encoder based on the ViT-Huge backbone. Swin Transformer (Liu et al., 2021) is a more recent contribution in computer vision. Unlike ViT, this architecture does not operate on fixed-size patches, as it starts from small patches and merges neighboring patches in deeper layers. We experiment with the large variant, Swin-L. On top of these deep models, we train a meta-regressor based on domain-adaptive kernel learning (DAKL). The meta-regressor combines the embeddings given by the deep architectures.

**Latent space model.** The U-Net in Stable Diffusion is a white-box model that operates in the latent space where the diffusion process takes place. The model is based on a three-stage approach. First, we use the perceptual compression auto-encoder used in Stable Diffusion to encode the images in the latent space. Next, we use an image captioning model (Li et al., 2022) to generate natural language descriptions. Lastly, based on the latent vectors and the captions, we train the encoder part of the U-Net architecture on our task, using the objective defined in Eq. (4). We use the captions similar to how the text is used for text-to-image generation, namely we integrate them in the cross-attention blocks of the U-Net.

# 4 EXPERIMENTS

**Data set.** DiffusionDB (Wang et al., 2022) contains 14 million images generated from 1.8 million prompts by Stable Diffusion (Rombach et al., 2022). However, we observed that many prompts are near duplicates, and keeping all of them could make our task very easy (models could cheat by memorizing samples). To clean up the prompts, we introduce a set of successive filtering steps: $(i)$ eliminating any leading or trailing whitespaces, $(ii)$ removing examples having less than one word or containing *Null* or *NaN* characters, $(iii)$ removing examples containing non-English characters, and $(iv)$ discarding duplicate prompts having at least 50 identical characters at the beginning or the end. After applying the proposed filtering steps, we obtain a clean data set of approximately 730,000 image-prompt pairs. We divide the pairs into 670,000 for training, 30,000 for validation and 30,000 for testing.

**Hyperparameters.** We provide details about hyperparameters in the supplementary.

**Results.** In Table 1, we report the average cosine similarity scores between the prompt embeddings predicted by various neural models and the ground-truth prompt embeddings in our test set. We compare vanilla models with versions that are enhanced by our novel training pipeline. We introduce our novel components one by one, to illustrate the benefit of each added component. We further report results with a joint framework based on the weighted average of the prompt embeddings generated by the four individual models. Another version based on all models is integrated with our DAKL method.

As shown in Table 1, our findings indicate that the classification head leads to consistent performance gains. For three out of four models, our curriculum learning strategy brings further performance improvements. Moreover, the joint pipeline obtains much better results, with a slight performance edge from the version based on DAKL. We thus conclude that all our novel contributions boost performance. Notably, our highest gains are observed for the U-Net$_{\text{enc}}$ model, as the performance grows from $0.6130$ to $0.6497$.

**Ablation study.** In Table 2, we present the results obtained by each head configuration illustrated in Figure 3. We observe that the third option gives the best results, showing

Table 1: Average cosine similarity scores between predicted and ground-truth prompt embeddings, employing different neural architectures, while gradually adding our novel components, namely the vocabulary classification head and the curriculum learning procedure, to illustrate their benefits. We also report the results of the joint framework, before and after integrating our DAKL method.

| Image Encoder | Multi-label classification | Curriculum learning | DAKL | Cosine similarity |
|---|---|---|---|---|
| CLIP-Huge | - | - | - | 0.6725 |
| CLIP-Huge (ours) | ✓ | - | - | 0.6739 |
| CLIP-Huge (ours) | ✓ | ✓ | - | **0.6750** |
| U-Net$_{enc}$ | - | - | - | 0.6130 |
| U-Net$_{enc}$ (ours) | ✓ | - | - | 0.6455 |
| U-Net$_{enc}$ (ours) | ✓ | ✓ | - | **0.6497** |
| Swin-L | - | - | - | 0.6624 |
| Swin-L (ours) | ✓ | - | - | **0.6671** |
| Swin-L (ours) | ✓ | ✓ | - | **0.6671** |
| ViT | - | - | - | 0.6526 |
| ViT (ours) | ✓ | - | - | 0.6539 |
| ViT (ours) | ✓ | ✓ | - | **0.6550** |
| All models | - | - | - | 0.6841 |
| All models (ours) | ✓ | - | - | 0.6879 |
| All models (ours) | ✓ | ✓ | - | 0.6900 |
| All models (ours) | ✓ | ✓ | ✓ | **0.6917** |

Table 2: Ablation study on the head configurations used for the classification task. The options illustrated in Figure 3 are tested on the Swin-L backbone.

| | Baseline | Head configuration | | |
|---|---|---|---|---|
| | | 1 | 2 | 3 |
| Cosine similarity | 0.6624 | 0.6620 | 0.6632 | **0.6671** |

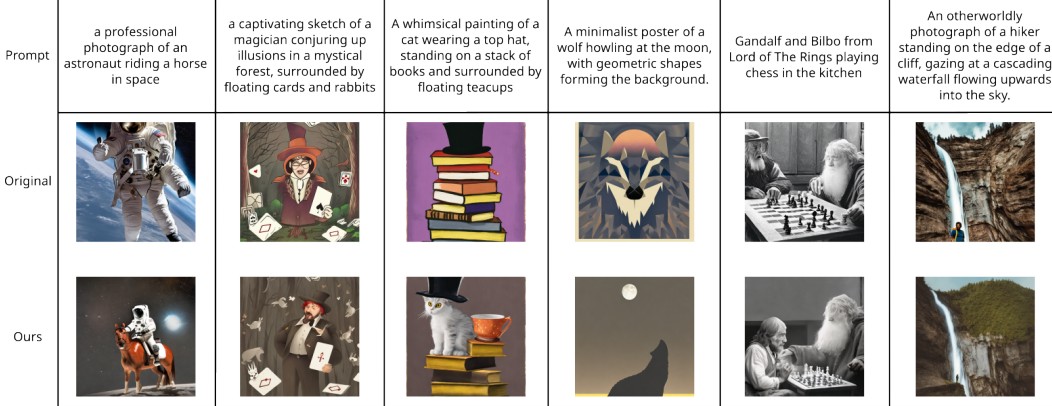

Figure 4: Samples generated by original and modified Stable Diffusion models. The images on the middle row are synthesized by the original U-Net. The images on the bottom row are generated by replacing (from the second diffusion step onward) the original U-Net encoder with our U-Net encoder employed for prompt embedding prediction. Note that the top image from the first column does not include a horse, while the bottom image contains one. A similar example is on the second column, where the rabbits are missing from the top image, but they are present in the bottom image. Another type of error corrected by our model is the appearance of people or objects. For example, the person in the image in the last column is looking away from the cascade, but the prompt specifies that the hiker should look towards the cascade. Another interesting example is on the fifth column, because our method recognizes that Bilbo is a hobbit, and he should therefore be shorter.

that it is useful to update the embedding head based on the classification error. This result strengthens the claim that our additional classification task is useful in estimating the sentence embeddings.

Table 3: Results of the subjective human evaluation study on image-text alignment completed by three volunteers. Each annotator was shown 100 image pairs and asked to choose the image that best aligns with the given prompt. For each displayed pair, the left or right image locations were randomly chosen to avoid cheating. Each person had a third option if they were unsure, denoted as *undecided*.

| Choice | Person #1 | Person #2 | Person #3 | Agreements | Average |
|---|---|---|---|---|---|
| Ours | 52 | 36 | 31 | 21 | 39.6 |
| Original | 30 | 10 | 5 | 2 | 15.0 |
| Undecided | 18 | 54 | 64 | 14 | 45.4 |

**Application to image generation.** We next demonstrate the promising capabilities of our framework. We specifically illustrate the ability to synthesize images that are better aligned with the prompts. This is achieved by simply replacing the U-Net in the Stable Diffusion pipeline with the one used in our prompt generation task. Our initial attempt resulted in spurious invalid images, *i.e.* very similar to noise. Therefore, for the first iteration only, we use the original U-Net, then switch to our U-Net for the remaining steps of the diffusion process. This approach produces more faithful visual representations of the prompts. Figure 4 depicts a few qualitative results, where our model exhibits better alignment between the generated image and the text prompt. We observe that the images generated by our model illustrate more details that are omitted in the first instance by the original U-Net.

To ensure a fair comparison of the original and modified Stable Diffusion models, we conducted an experiment based on human feedback. The experiment involved three individuals who evaluated 100 pairs of randomly sampled images, from an image-to-text alignment perspective. Each participant was given the same set of 100 image pairs. Each pair is formed of an image generated by our U-Net, and another one, generated by the original model. Both images in a pair are generated for the same prompt. The generative process ran 50 denoising steps with the DDIM sampler (Song et al., 2021a). The participants had to choose the image that is more representative for the given prompt, having three options: first image, second image, and undecided. A pair labeled as *undecided* denotes the case when a participant is not able to choose a better image. Moreover, the location of the images is assigned randomly within each pair to prevent disclosing the source model that generated each image, thus avoiding any form of cheating.

The detailed results of the above experiment are shown in Table 3. We report the number of votes awarded by each person to each of the two models, ours and the original, along with the number of undecided votes by each participant. Overall, the voting results show that, indeed, the usage of our U-Net in the generative process yields more text-representative images. Furthermore, we also compute the number of times the three participants agree on the same response, and we save the results in the fifth column. The agreement results are also favorable to our U-Net.

In summary, the outcome of the above experiment is remarkable. It is clear that our model performs better when evaluated from the text-to-image alignment perspective.

## 5 Conclusion

In this paper, we explored the task of recovering the embedding of a prompt used to generate an image, which represents the reverse process of the well-known text-to-image generation task. We proposed a joint training pipeline that integrates three novel components, and showed how they contribute to a higher performance. These are: a multi-label classification head for the most frequent words, a curriculum learning scheme for improved convergence, and a domain-adaptive kernel learning framework for meta-regression. In our experiments, we leveraged the large number of image-prompt pairs obtained from the DiffusionDB data set. We also demonstrated how a model trained on the image-to-text task can be beneficial to the original task, generating images that are more faithful to the input text prompts.

Learning to predict the original text prompts of diffusion models employed for text-to-image generation is a stepping stone towards gaining a deeper comprehension of these models. Through our research, we open a new path towards the development of more advanced and effective techniques in the area of diffusion modeling.

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
