# OpenReview forum: "Reverse Stable Diffusion: What prompt was used to generate this image?"
_ICLR.cc/2024/Conference — ICLR 2024 Conference Withdrawn Submission_

### Official Review · Reviewer_xLb4 · 2023-10-31

**Soundness:** 3 good
**Presentation:** 3 good
**Contribution:** 3 good
**Rating:** 6
**Confidence:** 3

**Summary:**

The paper proposes a framework to predict text-embedding based on the generated image from a Stable Diffusion. More precisely, it involves multiple components: a prompt-embedding and word classification training objective, a curriculum learning schedule, and a domain-adaptive kernel learning method to boost the performance. The paper also shows that the learned prompt-embeddings along with the images can be fed back for generative model training, which improves the model's text-image alignment.

**Strengths:**

1. The paper aims for an interesting task of predicting prompt embedding from Stable Diffusion generated images, and proposes multiple novel components to achieve the task.
2. The generated prompt-embeddings can later be paired with the images to fine-tune the generative model, which enhances the text-image alignment of the model.

**Weaknesses:**

1. I am generally questionable about the portion of domain-adaptive kernel learning:
    - Is it really applicable? From the setup, it seems to require  knowledge about the test domain. However, in real world task, the test domain could be unknown or infinitely large due to the generative power of a diffusion model. How should we define the test domain for this learning task?
    - It requires a hyper-parameter $r$ to define the number of k-means centroids. However, this $r$ is not studied and how should we choose it?

2. The paper has some typos. For example, on the top paragraph of Page 6, there's a "Figure ??" typo.

**Questions:**

Please refer to the weakness section.

---

> ### Author Response · Authors · 2023-11-21
> **Rebuttal letter**
>
> - _I am generally questionable about the portion of domain-adaptive kernel learning: Is it really applicable? From the setup, it seems to require knowledge about the test domain. However, in real world task, the test domain could be unknown or infinitely large due to the generative power of a diffusion model. How should we define the test domain for this learning task?_
>
> **Reply:** The proposed model requires some unlabeled data from the target domain. In our case, we consider that the validation and test domains are both from the target domain. Hence, in the DAKL method, we used the validation data. Still, the distribution gap between training and validation / test data is not significant. This explains why the performance gain given by DAKL is moderate.
>
> - _It requires a hyper-parameter r to define the number of k-means centroids. However, this is not studied and how should we choose it?_
>
> **Reply:** We have already conducted experiments with various values for r. Please see Table 3 from the supplementary material.
>
> - _The paper has some typos. For example, on the top paragraph of Page 6, there's a "Figure ??" typo._
>
> **Reply:** The respective figure was moved from the main paper to the supplementary, at the last moment. The text refers to Figure 4 from the supplementary. We uploaded a corrected version of the manuscript.

---

### Official Review · Reviewer_zz1f · 2023-11-01

**Soundness:** 2 fair
**Presentation:** 2 fair
**Contribution:** 2 fair
**Rating:** 3
**Confidence:** 4

**Summary:**

This paper proposes a framework to predict the original text-prompt embedding given the generated images by a text-conditioned diffusion model, in purpose of tuning the diffusion model toward better semantic alignment. The proposed framework employs an ensemble of four pre-trained visual-encoder backbones and trains a head on top of them to predict the original text-prompt embedding of BERT. One of the visual-encoder backbone is the U-Net of the pre-trained diffusion model, and as a by-product of the fine-tuning for the prompt-embedding prediction, it is claimed that the updated U-Net is better at generating the image with semantic alignment.

**Strengths:**

The considered topic of fine-tuning the text-to-image diffusion model toward better semantic alignment is important and of great interest. The proposed method seems to be simple.

**Weaknesses:**

- The main objective is unclear.
    - A large body of the manuscript and the title itself focus on predicting the original text embedding. However, as long as the actual caption is not generated but only the text embedding is predicted, it is unclear what the point is. As for evaluating the text-alignment, most of work already report CLIP similarity score, and the motive for making a new prediction model for text embedding seems to be weak. As for improving the U-net backbone via fine-tuning, one could also consult CLIP similarity loss first.
- The proposed method seems to be rather hand-wavy.
    - Why it has to be four encoder backbones, and why this specific combination (U-Net, ViT, Swin-T, CLIP) is needed? If this is solely for the performance, proper ablation study (using a subset of the backbones) is needed.
    - Why is it designed to predict the BERT embedding? CLIP-text encoder seems more welcoming, considering CLIP-image encoder is already used as a backbone, and another backbone, the diffusion U-Net, is highly related to it.
    - Does making a better prediction via extra techniques (Multi-label classification, Curriculum learning, DAKL; Table 1) necessarily lead to a better performance in image generation with the diffusion model? Although Fig. 4 shows the overall improvement, it is unclear if this is closely correlated to cosine-similarity score in Table 1.
- Comparison with other work is missing.
    - There are some recent studies proposing other fine-tuning methods for diffusion models (e.g., [1], [2]). The paper lacks comparison with these approaches.
        - [1] Aligning Text-to-Image Models using Human Feedback, Lee et al. 2023
        - [2] Emu: Enhancing Image Generation Models Using Photogenic Needles in a Haystack, Dai et al., 2023
    - While predicting the text-embedding proposed in this paper can be new, it should be compared with the captioning models as well. For example, by captioning the generated images with BLIP-2 and embedding the generated caption with BERT, the performance can be compared.
    ****
- Minor:
    - “Figure ??” on Page 6, the number is missing.
    - It is not entirely sure, but the Latex style feels a little off from the standard. The authors might want to double-check this.

---

The authors' rebuttal partly resolved my concerns and I updated the scores.

**Questions:**

- In vocabulary classification, it seems the training would mislead the model if there are synonyms or paraphrase. How is this handled?
- If the text-embedding cosine-similarity is evaluated with the newly generated images by the updated U-Net, do the scores get better?

---

> ### Author Response · Authors · 2023-11-21
> **Rebuttal letter**
>
> - _As long as the actual caption is not generated but only the text embedding is predicted, it is unclear what the point is._
>
> **Reply:** Although predicting the actual prompt is possible, we consider that this form of evaluating the task is ill-posed. This is because we have only one prompt for each generated image, while image captioning benchmarks typically have several alternative ground-truth captions for each image, and models are evaluated against the best matching ground-truth caption. Hence, predicting the exact prompt is significantly more difficult than predicting the prompt embedding. In the embedding space, we essentially allow the models to predict semantically related prompts (paraphrases) without being needlessly penalized. Moreover, this allows us to have a more objective evaluation, avoiding the use of multiple (sometimes suboptimal) measures to quantify the quality of the generated text prompts. This explains why directly predicting the text embeddings represents a more reasonable task. Nevertheless, we already performed image captioning experiments and showed the corresponding results in Table 1 and Figure 3 from the supplementary. Notice that our novel components, namely the multi-class head and the curriculum learning regime, boost the image captioning performance of BLIP. This shows that our contributions are suitable for both text generation and text embedding prediction from generated images.
>
> - _As for evaluating the text-alignment, most of work already report CLIP similarity score, and the motive for making a new prediction model for text embedding seems to be weak._
>
> **Reply:** As recommended, we computed the CLIP similarity score for the U-net model on our test data. The resulting score is 0.3287, which is well below the score of our models trained on the BERT embedding space. This result justifies the need to train models on the proposed task, i.e. predicting text embeddings of generated images.
>
> - _As for improving the U-net backbone via fine-tuning, one could also consult CLIP similarity loss first._
>
> **Reply:** U-net is already conditioned on CLIP embeddings, and the original Stable Diffusion jointly optimizes U-net and CLIP. Still, our results presented in Table 3 and Figure 4 from the main paper show that this is not enough. We believe that the use of a BERT-style encoder for the text embeddings brings something new to the table, and this helps U-net in learning a better alignment.
>
> - _Why it has to be four encoder backbones, and why this specific combination (U-Net, ViT, Swin-T, CLIP)?_
>
> **Reply:** We agree that our ensemble-based approach needs to be better motivated. Since the goal is to assess how well the text embedding can be recovered from generated images, we motivate our use of an ensemble of multiple models via the focus on minimizing the possibility of reporting low cosine similarity scores due to a poor model choice for the reverse task. In Table 1 from the main paper, there is a noticeable gap between the individual models and the ensemble. We thus believe the scores reported for the ensemble better reflect the misalignment of the original Stable Diffusion model. In the revised supplementary, we added new results in Table 5 with combinations of two and three models, which clearly indicate that all individual models play a role in the proposed ensemble (as discussed in Section 9 from the revised supplementary).
>
> - _Why predict the BERT embedding? CLIP-text encoder seems more welcoming, considering CLIP-image encoder is already used as a backbone, and another backbone, the diffusion U-Net, is highly related to it._
>
> **Reply:** According to [i], BERT-style encoders are better than CLIP-style encoders, when it comes to encoding text. We thus consider that for our text prediction task, the sentence BERT encoder generates more representative embeddings. We clarified this aspect in the revised manuscript.
>
> [i] _Zhihong Chen, Guiming Chen, Shizhe Diao, Xiang Wan, and Benyou Wang. 2023. On the Difference of BERT-style and CLIP-style Text Encoders. In Findings of the Association for Computational Linguistics: ACL 2023, pages 13710–13721. https://aclanthology.org/2023.findings-acl.866.pdf_
>
> - _Does making a better prediction via extra techniques (Multi-label classification, Curriculum learning, DAKL) necessarily lead to a better performance in image generation with the diffusion model?_
>
> **Reply:** To answer this question, we generated several images with the fine-tuned U-Net, before and after introducing our extra techniques. The new images are included in Figure 2 from the revised supplementary. There are several cases where the enhanced version of U-Net produces better aligned images (as discussed in Section 2 from the revised supplementary). Moreover, we applied our extra techniques in image captioning, on the BLIP model. The results presented in Table 1 and Figure 3 from the supplementary show that the extra techniques bring performance gains for BLIP as well.

---

> > ### Comment · Reviewer_zz1f · 2023-11-22
> > **Reply to the rebuttal**
> >
> > > U-net is already conditioned on CLIP embeddings, and the original Stable Diffusion jointly optimizes U-net and CLIP. Still, our results presented in Table 3 and Figure 4 from the main paper show that this is not enough. We believe that the use of a BERT-style encoder for the text embeddings brings something new to the table, and this helps U-net in learning a better alignment.
> >
> > > According to [i], BERT-style encoders are better than CLIP-style encoders, when it comes to encoding text. We thus consider that for our text prediction task, the sentence BERT encoder generates more representative embeddings. We clarified this aspect in the revised manuscript.
> >
> > * Stable Diffusion does not optimize the CLIP model.
> > * Table 3 and Figure 4 just show the original model results. What I meant is that the authors' framework can be done for the CLIP model. That is, instead of taking the cos-similarity loss with the BERT embedding, one could take a cos-similarity loss with the CLIP embedding, which seems to be more straightforward as the U-Net is conditioned on CLIP. Currently, it is unclear that whether the improved semantic alignment of the generated images is due to the proposed embedding prediction framework itself or due to the newly introduced BERT model.
> >
> > > We agree that our ensemble-based approach needs to be better motivated. ...
> >
> > > In the revised supplementary, we added new results in Table 5 with combinations of two and three models, which clearly indicate that all individual models play a role in the proposed ensemble
> >
> > * An ensemble model is, of course, expected to perform better than a single, but the question here is that whether the improved prediction actually leads to the better fine-tuning of the U-Net (in terms of semantic alignment). My concerns are:
> >   * Other backbone models in the ensemble (ViT, Swin-T, CLIP) do not directly affect the fine-tuning of the U-Net.
> >   * Accurately predicting the BERT embedding does not neccessarily mean it becomes a better evaluation/training criterion. For example, if the ranking between the models (e.g., the original model and the proposed fine-tuned model) is unchanged or if the text-retrieval results using the predicted embedding are unchanged, ensemble might not be needed.
> > * It is not that improving the BERT prediciton is unimportant but that its necessity seems to be not clrealy presented.
> >
> > Other replies resolve my questions and concerns.

---

> > > ### Author Response · Authors · 2023-11-22
> > > **Reply to reply :)**
> > >
> > > We thank the reviewer for reading our rebuttal. We would like to provide additional clarifications below.
> > >
> > > - _Stable Diffusion does not optimize the CLIP model._
> > >
> > > **Reply:** According to the Stable Diffusion paper [ii], the authors mention at page 5 that "both $\tau_\theta$ and $\epsilon_\theta$ are jointly optimized via Eq. 3". As per. Fig. 3 from [ii], please note that $\tau_\theta$ is the CLIP text encoder and $\epsilon_\theta$ is the U-Net. We are sorry for not making this more clear in our rebuttal letter. We hope this point is now clarified.
> > >
> > > [ii] https://arxiv.org/abs/2112.10752
> > >
> > > - _What I meant is that the authors' framework can be done for the CLIP model._
> > >
> > > **Reply:** As suggested, we fine-tuned the U-Net model on the CLIP embeddings of the prompts. The model converges very fast (in about 1 epoch) to 0.92 cosine similarity, as it starts with a similarity of 0.89. This shows that the model is already well adapted on the CLIP space. However, as also mentioned in ref. [i], the CLIP space is not ideal for text representation. Ref. [i] shows that BERT space is more suitable for representing text. Since our goal is to design a task that closely resembles image captioning, we consider that the BERT space is more suitable for our image-to-prompt prediction task than the CLIP space.
> > >
> > > - _Other backbone models in the ensemble (ViT, Swin-T, CLIP) do not directly affect the fine-tuning of the U-Net._
> > >
> > > **Reply:** Yes, this is correct. The ensemble is designed for the text embedding prediction task. The generation of images that are better aligned with the captions via the fine-tuned U-Net is a byproduct / application of the fine-tuning process applied to U-Net.
> > >
> > > - _Accurately predicting the BERT embedding does not neccessarily mean it becomes a better evaluation/training criterion. For example, if the ranking between the models (e.g., the original model and the proposed fine-tuned model) is unchanged or if the text-retrieval results using the predicted embedding are unchanged, ensemble might not be needed._
> > >
> > > **Reply:** In Table 1 from the supplementary, we showed image captioning results with a 1-NN model based on the proposed ensemble. The model achieves a RefCLIPScore of 25.53. We tested the best individual model from the ensemble, namely CLIP, with the 1-NN approach and we obtained a RefCLIPScore of 24.09. This confirms that the ensemble is able to retrieve better captions than the CLIP model.
> > >
> > > - _It is not that improving the BERT prediction is unimportant but that its necessity seems to be not clearly presented._
> > >
> > > **Reply:** We agree that this point was not made sufficiently clear. We would like to underline that our original aim is to reverse the text-to-image generation task. However, predicting the actual text prompts is problematic, since paraphrases of the ground-truth captions would receive low scores using common evaluation metrics, e.g. BLEU. The BERT space allows us to score paraphrases and, according to ref. [i], it is a more suitable representation for text than the CLIP space. CLIP is suitable to jointly represent image and text samples in a single space, but our main objective is to accurately score paraphrases of the target sentence.
> > >
> > > Of course, one can argue that the models could be tasked at generating captions and we could apply the BERT encoder on top of captions. This was done with BLIP in Table 4 from the supplementary, and the results show that this approach underperforms. Hence, we strongly believe that optimizing the models to directly predict the BERT embeddings of captions is the better choice.

---

> ### Author Response · Authors · 2023-11-21
> **Rebuttal letter (continued)**
>
> - _There are some recent studies proposing other fine-tuning methods for diffusion models (e.g., [1], [2]). The paper lacks comparison with these approaches._
>
> **Reply:** We thank the reviewer for finding these related works. To the best of our knowledge, we are the first to reverse text-to-image diffusion. The suggested models are designed for a particular application of our method, but our model is more generic. Since our main objective is not particularly image generation, we do not consider that [1] and [2] are not closely related. Moreover, the suggested models for image generation are much heavier and complex, even requiring human feedback during training, e.g. [1]. Since human feedback for Diffusion DB is not available, the comparison with [1] is not directly achievable. Furthermore, we note that [2] is a preprint published on September 27, 2023 (one day before the ICLR submission deadline), so the comparison with [2] would have been impossible in one day, especially because there is no public code available for [2].
>
> - _While predicting the text-embedding proposed in this paper can be new, it should be compared with the captioning models as well. For example, by captioning the generated images with BLIP-2 and embedding the generated caption with BERT, the performance can be compared._
>
> **Reply:** In Table 4 from the supplementary, we already show results with the fine-tuned BLIP image captioning model. The generated caption is embedded using the same sentence encoder as for the ground-truth prompts. This kind of approach obtains suboptimal results, since BLIP obtains a cosine similarity of 0.51, while the models based on directly predicting the embeddings are all above 0.6. Moreover, we present image captioning results in Table 1 in the supplementary. Notice that our pipeline based on multi-label classification and curriculum learning leads to better captioning results for BLIP.
>
> - _“Figure ??” on Page 6, the number is missing._
>
> **Reply:** The respective figure was moved from the main paper to the supplementary, at the last moment. The text refers to Figure 4 from the supplementary. We uploaded a corrected version of the manuscript.
>
> - _It is not entirely sure, but the Latex style feels a little off from the standard. The authors might want to double-check this._
>
> **Reply:** We double checked the version of the Latex style file and confirm it is the one downloaded from the ICLR 2024 website.
>
> - _In vocabulary classification, it seems the training would mislead the model if there are synonyms or paraphrase. How is this handled?_
>
> **Reply:** The proposed head performs multi-label classification, i.e. the target is NOT a one-hot vector. When there are two synonyms, the model would simply have to predict which of the two synonyms is used. The multi-label classification head requires learning specific words and distinguishing between synonyms, but this is not misleading the model. A quick analysis of the words (based on WordNet) shows that less than 2% are synonyms.
>
> - _If the text-embedding cosine-similarity is evaluated with the newly generated images by the updated U-Net, do the scores get better?_
>
> **Reply:** As suggested, we generated a new set of images for 2500 randomly chosen test prompts with the updated U-Net model. Then, we employed the model to predict the text embeddings. The resulting similarity for the 2500 samples grows from 0.6417 to 0.6443, confirming that the new U-Net generates more aligned images. We thank the reviewer for this suggestion.

---

### Official Review · Reviewer_2mcu · 2023-11-02

**Soundness:** 2 fair
**Presentation:** 3 good
**Contribution:** 2 fair
**Rating:** 5
**Confidence:** 4

**Summary:**

This paper attempts to enhance the understanding of stable diffusion by leveraging predictions on prompt embeddings, while simultaneously improving the capability of stable diffusion. The paper's completeness is fair, and the experiments are comprehensive. However, the motivation behind the paper is not clear, the method lacks novelty, and the overall contribution is limited.

**Strengths:**

The paper is reasonably well-rounded in terms of completeness, the experiments are generally sufficient, and the figures and charts are quite clear.

**Weaknesses:**

The motivation in this paper is very weak, making it difficult to identify significant innovation or contributions. The motivation as I understand it in the paper is to enhance the understanding of the diffusion model through embeddings. However, the method section only discusses how to obtain embeddings, without addressing how this enhances the understanding of the diffusion model. If the motivation is just predict the text embeddings, Image Caption + CLIP can already be achieved. Furthermore, I see significant contradictions in its methodology. In the paper, it is mentioned that the model itself has semantic issues. However, it suggests that training a model using these problematic images generated by the model can actually improve the issue, which, in my opinion, is contradictory.

**Questions:**

Question:

1. It seems that your method might not fulfill your motivation. Your motivation aims to enhance the understanding of the diffusion model, but your method primarily focuses on predicting prompt embeddings. How does this directly contribute to a better understanding of the diffusion model?

2. Regarding the method, you later mention using the model you trained to participate in generating and claim that this can address some issues in the original model's prompt understanding, such as object omissions. However, the data used for training your model consists of images with issues generated by the original model. How does a model trained on problematic data contribute to resolving the issues in the original model?

3. In terms of novelty, what are the distinct advantages of computing embeddings using your method compared to directly using an image caption model to predict prompts and then calculating embeddings using an image encoder?

---

> ### Author Response · Authors · 2023-11-21
> **Rebuttal letter**
>
> - _The motivation in this paper is very weak, making it difficult to identify significant innovation or contributions. The motivation as I understand it in the paper is to enhance the understanding of the diffusion model through embeddings. However, the method section only discusses how to obtain embeddings, without addressing how this enhances the understanding of the diffusion model.
> It seems that your method might not fulfill your motivation. Your motivation aims to enhance the understanding of the diffusion model, but your method primarily focuses on predicting prompt embeddings. How does this directly contribute to a better understanding of the diffusion model?_
>
> **Reply:** Our contribution is a just step towards understanding of diffusion models, but we do agree that there are many things to be addressed in order to completely solve the understanding of diffusion models. However, our study contributes to the better understanding and use of diffusion models in several ways. First, our study aims to elucidate if one can reverse the text-to-image diffusion process. To address this task, we developed a powerful ensemble based on several models. The cosine similarity values present in Table 1 range between 0.65 and 0.7, indicating that the text-to-image diffusion process can be reversed, only by some degree. The results indicate that, in the original diffusion model, there is a misalignment between the original text prompts and the generated images. Hence, our study shows a way to quantify the misalignment gap. Second, as shown in our first application, the proposed method can be used to generate images that are much better aligned with the text prompt.
>
> - _Furthermore, I see significant contradictions in its methodology. In the paper, it is mentioned that the model itself has semantic issues. However, it suggests that training a model using these problematic images generated by the model can actually improve the issue, which, in my opinion, is contradictory. Regarding the method, you later mention using the model you trained to participate in generating and claim that this can address some issues in the original model's prompt understanding, such as object omissions. However, the data used for training your model consists of images with issues generated by the original model. How does a model trained on problematic data contribute to resolving the issues in the original model?_
>
> **Reply:** While the input images are indeed generated, our model is trained to predict the human text prompts. In order to generalize well on the proposed task, the model learns to match images that are well aligned with their prompts. Hence, during the training process, the model focuses on images that are consistent with their prompts. This explains why the model is able to generate images that are better aligned with the input prompts. We present quantitative and qualitative results to support our claim. Moreover, we show that the U-Net encoder, which we trained on generated images to predict the prompt embedding, performs better at synthesizing images than the original U-Net in Stable Diffusion.
>
> - _If the motivation is just predict the text embeddings, Image Caption + CLIP can already be achieved. In terms of novelty, what are the distinct advantages of computing embeddings using your method compared to directly using an image caption model to predict prompts and then calculating embeddings using an image encoder?_
>
> **Reply:** In Table 4 from the supplementary, we already show results with the fine-tuned BLIP image captioning model. The generated caption is embedded using the same sentence encoder as for the ground-truth prompts. This kind of approach obtains suboptimal results, since BLIP obtains a cosine similarity of 0.51, while the models based on directly predicting the embeddings are all above 0.6. Although predicting the actual prompt is possible, we consider this task ill-posed. This is because we have only one prompt for each generated image, while image captioning benchmarks typically have several alternative ground-truth captions for each image, and models are evaluated against the best matching ground-truth caption. Hence, predicting the exact prompt is significantly more difficult than predicting the prompt embedding (which matches paraphrases more easily). In the embedding space, we essentially allow the models to predict semantically related prompts without being needlessly penalized. This explains why directly predicting the text embeddings leads to better results.